# Quality Appraisal of Nutritional Guidelines to Prevent, Diagnose, and Treat Malnutrition in All Its Forms during Pregnancy

**DOI:** 10.3390/nu14214579

**Published:** 2022-11-01

**Authors:** Cinthya Muñoz-Manrique, Mónica Ancira-Moreno, Soraya Burrola-Méndez, Isabel Omaña-Guzmán, Elizabeth Hoyos-Loya, Sonia Hernández-Cordero, Alejandra Trejo-Domínguez, Mónica Mazariegos, Natalia Smith, Scarlett Alonso-Carmona, Jennifer Mier-Cabrera, Loredana Tavano-Colaizzi, Belén Sánchez-Múzquiz, Fermín Avendaño-Álvarez, Karla Muciño-Sandoval, Nadia C. Rodríguez-Moguel, Magali Padilla-Camacho, Salvador Espino-y-Sosa, Lizeth Ibarra-González, Cristina Medina-Avilés

**Affiliations:** 1Health Department, Universidad Iberoamericana, Prolongación Paseo de Reforma 880, Lomas de Santa Fe, Mexico City 01219, Mexico; 2Maternal and Child Health and Nutrition Network (MaCHiNNe), Observatorio Materno Infantil (OMI), Universidad Iberoamericana, Prolongación Paseo de Reforma 880, Lomas de Santa Fe, Mexico City 01219, Mexico; 3Instituto Nacional de Perinatología, Montes Urales 800 Col., Virreyes Del., Miguel Hidalgo, Mexico City 11000, Mexico; 4Research Center for Equitable Development EQUIDE, Universidad Iberoamericana, Prolongación Paseo de Reforma 880, Lomas de Santa Fe, Mexico City 01219, Mexico; 5Research Center for the Prevention of Chronic Diseases (CIIPEC), Institute of Nutrition of Central America and Panama (INCAP), Calzada Roosevelt, 6.25 Zona 11, Guatemala City 01011, Guatemala; 6Department of Pediatrics, University of California, San Francisco, San Francisco, CA 94143, USA; 7Departamento de Investigación en Enfermedades Infecciosas, Instituto Nacional de Enfermedades Respiratorias, Calzada de Tlalpan 4502, Belisario Domínguez Secc 16, Tlalpan, Mexico City 14080, Mexico; 8Banco de Leche Humana del Estado de Guanajuato, Vialidad Interior Malvas 367, Revolución, Irapuato 36546, Mexico

**Keywords:** AGREE II, clinical practice guidelines, pregnancy, healthy pregnancy, malnutrition

## Abstract

This work aimed to identify clinical practice guidelines (CPGs) that include recommendations for the prevention, diagnosis, and treatment of women’s malnutrition during pregnancy and to evaluate the quality of these guidelines using the Appraisal of Guidelines for Research and Evaluation (AGREE II) instrument. We conducted a literature review using PubMed and different websites from January 2009 to February 2021. The quality of the CPGs was independently assessed by reviewers using the AGREE II instrument, which defines guidelines scoring >70% in the overall assessment as “high quality”. The analysis included 43 guidelines. Among the main findings, we identified that only half of the CPGs (51.1%) obtained a final “high quality” evaluation. AGREE II results varied widely across domains and categories. The two domains that obtained the highest scores were scope and purpose with 88.3% (range 39 to 100%) and clarity of presentation with 87.2% (range 25 to 100%). Among the “high quality” CPGs, the best scores were achieved by the three guidelines published by the National Institute of Health and Care Excellence (NICE) and the World Health Organization (WHO). Due to the importance of maternal nutrition in pregnancy, it is essential to join forces to improve the quality of the guidelines, especially in CPGs that do not meet the reference standards for quality.

## 1. Introduction

Maternal nutrition crucially contributes to the offspring’s health and future generations. Studies in animals and humans have observed adverse developmental processes around conception in response to various environmental situations. The quality of the maternal diet, undernutrition, overweight, obesity, and physiological aspects related to glucose and lipid metabolism alteration may affect the embryo’s potential with consequences for risk of diseases in the short and long term [1].

Prepregnancy obesity is related to several adverse perinatal outcomes, with the risks increasing as the body mass index increases [2]. Regarding the newborn’s nutritional status, due to a proinflammatory status, metabolism alterations in preconception, and other factors, a woman with obesity may have a baby who is large for gestational age. These preconception conditions may also influence placental function and increase the risk of preeclampsia, intrauterine growth restriction, and gestational diabetes mellitus (GDM) [3]. Likewise, women with prepregnancy obesity have a higher probability of excessive gestational weight gain (GWG) and postpartum weight retention [4].

In the same way, the effect of maternal malnutrition is well known. Even though there is a smaller proportion of underweight women worldwide, these proportions are higher in low- and middle-income countries. Women with prepregnancy underweight have a greater probability of having a baby with inadequate fetal growth. These phenotypes may have a detrimental impact on children in their forthcoming years of schooling, educational performance, and health [5].

Women with micronutrient deficiencies before pregnancy, such as iron, folate, iodine, zinc, selenium, and vitamin B complex, are at a greater risk of having these deficiencies during pregnancy as their micronutrient requirements increase to promote adequate fetal development and growth. Deficiencies in critical micronutrients throughout the periconceptional period may affect maternal and offspring health, resulting in early reproductive failure, poor placental vascularization, inadequate fetal brain development, and poor fetal growth. Poor-quality diet intake or altered absorption due to infection, diseases, or inflammation could explain these nutrient deficiencies [6].

Conditions such as inadequate GWG, GDM, and hypertension disorders during pregnancy might influence fetal growth and women’s health. Systematic reviews have shown promising results in promoting adequate GWG and glycemic control by following nutrition and lifestyle recommendations during pregnancy [7,8]. In addition, observational studies have reported positive results in preventing hypertensive disorders, low birth weight, and preterm birth in women with better diet quality indices compared to women with lower dietary quality scores [9,10,11].

The problems of malnutrition mentioned above highlight the need to offer pregnant women food, nutrition, and lifestyle guidelines. International organizations prioritize maternal nutrition in the national public health plan to support a healthy pregnancy. However, women’s adherence to nutrition guidelines during pregnancy appears to be below the recommended level [12]. Several sociodemographic factors may influence guideline adherence, although the quality of the recommendations could also affect adherence due to the lack of clear evidence for health professionals to issue these recommendations.

Clinical practice guidelines (CPGs) are proper instruments to implement evidence-based guidelines for patients. However, few studies have evaluated the quality of pregnancy guidelines using a specific methodology. This paper aims to identify CPGs that include recommendations for preventing, diagnosing, and treating women’s malnutrition during pregnancy and to evaluate the quality of these guidelines using the Appraisal of Guidelines for Research and Evaluation (AGREE II) instrument.

## 2. Materials and Methods

### 2.1. Study Design and Eligibility Criteria

We thoroughly searched CPGs containing nutritional and lifestyle recommendations to prevent, diagnose, and treat all forms of malnutrition during pregnancy. We followed the framework proposed by Arksey and O’Malley [13] and further developed by Levac et al. [14] and the Joanna Briggs Institute [15]. This review process consisted of five stages: (i) identifying the research question; (ii) identifying relevant studies; (iii) study selection; (iv) charting the data; and (v) collating, summarizing, and reporting results. We added an extra step of (vi) critical appraisal using the AGREE II instrument [16].

The inclusion criteria considered international and national CPGs, consensus, expert opinion based on evidence, and standard references with recommendations on nutritional assessment (behavior, anthropometric, biochemical, clinical evaluation, and lifestyle), healthy diet, dietary modifications, nutritional supplementation, or any nutritional or lifestyle recommendation given in primary healthcare facilities. We excluded opinions, editorials, articles published as communication tools, and CPGs with lifestyle and nutrition recommendations oriented to a specific pathology or associated complications.

### 2.2. Search Strategy and Studies Selection

We performed a systematic literature search in the database PubMed (https://ncbi.nlm.nih.gov/pubmed, accessed on 10 February 2021) and a manual search of guideline-related websites, limiting the search to studies published between January 2009 and February 2021. The selection of this time frame considered the review and update of the Institute of Medicine (IOM) recommendations for weight gain during pregnancy that included ways to encourage their adoption through strategies to assist practitioners with the publication of the “Weight Gain During Pregnancy: Reexamining the Guidelines” in 2009 [17].

For the PubMed search, we used keywords such as pregnancy; pregnancy trimester, first; nutrition assessment; prevention and control; and malnutrition. We followed a Boolean search using search filters to identify any guidelines in English or Spanish (see Table 1). For the guideline-related websites, we selected international agencies and gynecology, obstetrics, and maternal nutrition societies as well as databases from the National Institute of Health and Care Excellence (NICE; UK) and Guidelines International Network (GIN). Key terms of the algorithm were used alone or in combination. We imported all studies identified through the database and the website search to Excel. We checked for and removed duplicates.

### 2.3. Quality Assessment

Authors, including dietitians and physicians, participated in the evaluation process. Two authors (CMM and MAM) independently assessed each study’s title and abstract to determine the references’ eligibility. In case of disagreement, another author (SBM) evaluated the guide. Several health professionals (CMA, GAC, JMC, KMS, LIG, LTC, MAM, MPC, NSG, SES, AT, BSM, CMM, FAA, and SHC) conducted a peer review to assess the full text of the potentially eligible documents in order to determine if they meet the inclusion criteria. In case of disagreement, a third author (AT, BSM, CMM, GAC, MPC, and SBM) was responsible to decide whether or not to include it.

We used the AGREE II tool to assess the quality of the selected CPGs. This instrument aims to identify the quality guidelines and their strengths and limitations and consists of 23 key items grouped into six domains: (1) scope and purpose (related to the overall aim of the guideline); (2) stakeholder involvement (measures the extent to which the appropriate stakeholders developed the guideline); (3) rigor of development (concerned with the process used to gather and synthesize the evidence); (4) clarity of presentation (appraises the language, structure, and format of the guideline); (5) applicability (related to the implications of applying the guideline); and (6) editorial independence (evaluates that the formulation of recommendations is unbiased with competing interests) followed by 2 global rating items (“overall assessment”). The evaluation of each item uses a 7-point Likert rating scale (from 1 = “strongly disagree” to 7 = “strongly agree), as defined in the AGREE II user’s manual [16].

The overall scores of each domain were calculated by adding their corresponding items and scaling the total as a proportion of the maximum possible score for that domain (max score = 100). The overall assessment requires the user to make a judgment as to the quality of the guideline, taking into account the criteria considered in the assessment process. The user is also asked whether he/she would recommend use of the guideline. A score of >70% indicates “high quality” in the guidelines as determined by the AGREE II user’s manual [16].

Due to time restrictions, only two authors (BSM, LTC, NRM, SES, AT, FAA, JMC, LIG, and MAM) independently evaluated the quality of each CPG using the online AGREE platform, “My AGREE PLUS”, which is suggested by the AGREE II methodology.

### 2.4. Data Analysis

Means and median scores were calculated in each AGREE II domain to identify the most critical domains across the different guidelines. The overall quality evaluation of each guideline used a threshold of 70% for the final score of each domain. We used Microsoft Excel 2021, version 16.57, for data collection and extraction.

## 3. Results

Figure 1 provides the PRISMA flow diagram [18] with a detailed summary of the research results. We identified 82 records through the search. After removing duplicated studies, the abstracts of 79 records were screened, leading to 54 CPGs for full review and, finally, 43 for quality assessment.

Table 2 shows the general characteristics of the analyzed CPGs. Regarding geographical distribution, 35% of the reviewed guidelines corresponded to the North America Region, with guidelines from Canada (n = 9) and the USA (n = 6), while 37% were from Europe and the Central Asia region, with contribution from the United Kingdom (n = 9), Poland (n = 2), Italy (n = 1), Spain (n = 1), France (n = 1), and Europe region (n = 2). The regions with the lowest number of CPGs reviewed were East Asia and the Pacific, with two CPGs from Australia and the Middle East and North Africa and one CPG from the United Arab Emirates. Finally, we included nine international CPGs (21%).

Concerning the type of recommendations in the reviewed CPGs, 86% had recommendations related to prevention, 56% corresponded to diagnosis, and 27% related to treatment (Table 2) of any form of malnutrition during pregnancy.

### 3.1. Quality of Guidelines According to the AGREE II Domains

Table 3 reports the total score (mean and median) for each domain and the final quality evaluation of all CPGs. Only 51.1% of CPGs obtained a final evaluation of “high quality” with four or more domains reaching a score higher than 70%. Among the “high quality” CPGs, “Antenatal care for uncomplicated pregnancies” (NICE, 2019) [44] had the best score with >90% in all domains, while “Guideline: daily iron supplementation in adult women and adolescent girls” (World Health Organization (WHO), 2016) [54] and “Guideline: vitamin A supplementation in pregnant women” (WHO,2011) [58] both scored >80% in all six domains.

On the other hand, a CPG published by Bomba-Opoń (2017) [23] and another one published by the American Dietetic Association (ADA), American Society of Nutrition (ASN), Siega-Riz, A.M., King, J.C. (2019) [21] obtained the lowest scores with 17 and 25% of total average, respectively.

When looking more thoroughly at domain scores, the highest domain scores were for “scope and purpose” (domain 1) with a mean score of 88.3% (range 39 to 100%) and “clarity of presentation” (domain 4) with a mean score of 87.2% (range 25 to 100%). Meanwhile, the lowest values were for “applicability” (domain 5) with a mean score of 67.8% (range 4 to 100%) and “rigor and development” (domain 3) with a mean score of 74.3% (range 10 to 100%).

Figure 2 and Figure 3 show the mean and median quality scores of each domain in the guidelines graded as high and low quality.

#### 3.1.1. Scope and Purpose Domain

Domain 1 had the highest results with a mean of 88.3% (range 39 to 100%) and a median of 89% (range 39 to 100%). Moreover, 30.2% (n = 13) of the guidelines had a maximum score of 100%. Only one guideline presented a score below 50%, which was the Bomba-Opoń (2017) guideline [23] with 39%. 

#### 3.1.2. Stakeholder Involvement

The mean score in domain 2 was 75.4% (range 28 to 100%), and the median was 83% (range 28 to 100%). Only 67.4% (n = 29) of the guidelines received scores higher than 70%. The guideline published by Maxwell, C. (2017) [41] and those by NICE in 2010 [45] and 2020 [46] had the highest scores (100% each). The Bomba-Opoń guideline [23] scored the lowest score of 28%.

#### 3.1.3. Rigor of Development

Regarding domain 3, the best score was 100% (range 10 to 100%), which was obtained by Maxwell, C. (2019) [40] and NICE (2019) [44]. In contrast, the lowest scores were 21 and 22% for the guidelines by ADA, ASN, Siega-Riz, A.M., and King, J.C. [21] and Bomba-Opoń [23], respectively.

#### 3.1.4. Clarity of Presentation

The mean score in domain 4 was the second highest with 87.2% (range 25 to 100%). Only 27.9% (n = 12) of the guidelines had a high score (100%). In contrast, the guidelines of Cetin, I. [24] and Donnay, S. [30] obtained the lowest scores of 25 and 42%, respectively.

#### 3.1.5. Applicability

Domain 5 had the worst result, with the lowest mean score of 67.8% (range 4 to 100%) and a median of 73% (range 4 to 100%). Only 51.1% (n = 22) of the reviewed guidelines obtained a score >70%, and 25.5% (n = 11) had a score below or equal to 50%. The WHO guidelines [55,57] received a high score of 100%. Once again, the guidelines by Bomba-Opoń [23] and ADA, ASN, Siega-Riz, A.M., and King, J.C. [21] obtained the lowest scores with 4 and 25%, respectively.

#### 3.1.6. Editorial Independence

For domain 6, the mean score was 75% (range 4 to 100%), and the median score was 92% (range 4 to 100%). Nine of the CPGs analyzed (20.9%) received a scores below or equal to 50%. Compared with the others, this domain obtained the lowest scores (e.g., Lausman, A. [38] obtained 4%, Ryan, K. [49] obtained 8%, and [23] also obtained 8%).

## 4. Discussion

The current study presents a rigorous review that explores the quality of 43 CPGs designed to prevent, diagnose, and treat malnutrition in pregnancy using the AGREE II tool. Among the main findings, we identified that only half of the CPGs (51.1%) obtained a final “high quality” evaluation with four or more domains reaching a score higher than 70%. The guidelines classified as high and low quality had a higher evaluation in the classifications of domain 1 “scope and purpose” (mean = 88.3%, range = 39–100%) and domain 4 “clarity of presentation” (main = 87.2%, range = 25–100%). The domains with the lowest score in both groups were “applicability” (mean = 67.8%, range = 4–100%), “rigor of development” (mean = 74.3.3%, range = 10–100%), and “editorial independence” (mean = 75.0%, range = 4–100%).

To our knowledge, there are few systematic reviews that have evaluated the quality of nutrition care guidelines in this population [62,63,64]. These systematic reviews focused mainly on the assessment of weight management guidelines. The findings published by all of them are consistent with ours. The quality of the guidelines regarding the “rigor of development” and “applicability” domains is poor. These observations impact the failure to establish recommendations in clinical practice and, consequently, to address malnutrition in all its forms in pregnant women. As international organizations recognize them, all the guidelines identified the objective and purpose of care in clinical practice. However, it is necessary to go beyond the purpose because clinical practice guidelines are developed to improve the medical decision-making process among health professionals [65]. Likewise, considering other nutrition indicators, such as food intake, diet quality, and nutrition biomarkers, to assess clinical nutrition guidelines may improve patient-centered antenatal care.

Pregnancy is a unique time for identifying, preventing, and correcting any form of malnutrition in women, which may impact the health of the next generation. This opportunity is of utmost importance as this is a sensitive time, and most countries integrate antenatal care into universal health access. Dietary interventions to promote a healthy diet during pregnancy are part of the double-duty actions suggested by a group of experts to end malnutrition in all its form by 2030 [66]. However, from a nutrition care approach, it is recommended that the nutrition care process be followed to establish specific and evidence-based dietary interventions for each person. In this context, researchers and professionals should strengthen the nutrition methodology in research studies and consider the best evidence-based clinical nutrition practice guidelines.

Assessment of the quality of CPGs is relevant in the clinical field and for formulating and implementing programs and interventions to improve the quality of care and the population’s nutritional status. We recommend that guideline developers, clinicians, researchers, and policymakers consider and use the AGREE II instrument as it is detailed and easy to apply in situations of poor nutrition in pregnant women and other clinical conditions [64]. In addition, this tool allows critical evaluation before implementation, which would support decision-making around the health system of a country or region and even during emergencies.

Maternal nutritional status during pregnancy plays a critical role in fetal growth and development, contributing to the offspring’s health and future maternal health; therefore, improving the quality of the domains with lower scores is essential. Enhancing the quality of CPGs could improve nutritional care during this life stage and the health of mothers and their offspring across generations. Due to the importance of maternal nutrition in pregnancy, it is essential to join forces to improve the quality of the guidelines, especially CPGs that need to meet the reference standards for quality.

## Figures and Tables

**Figure 1 nutrients-14-04579-f001:**
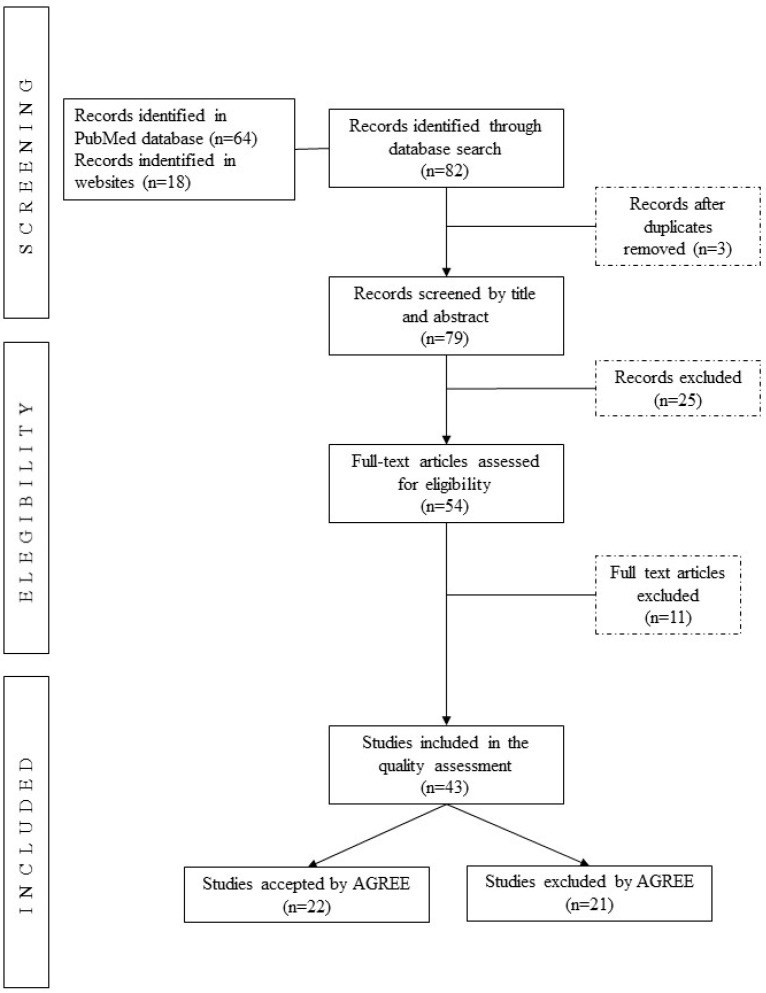
PRISMA flow diagram of searching and selecting guidelines.

**Figure 2 nutrients-14-04579-f002:**
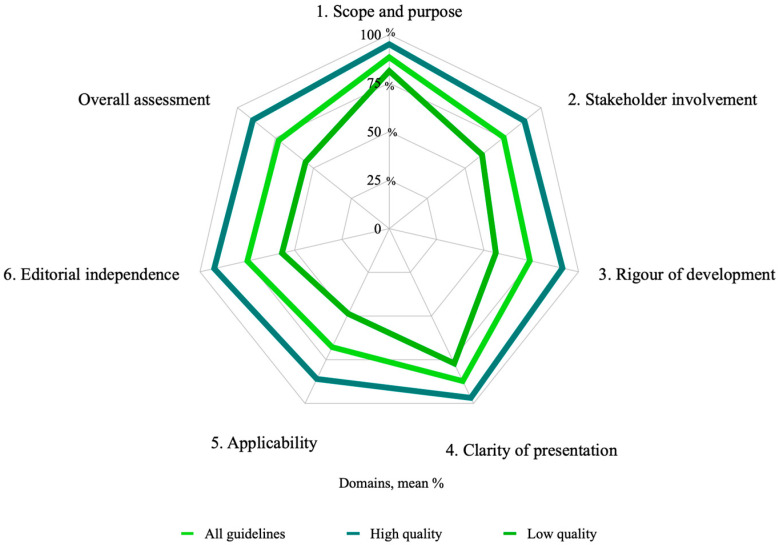
Average quality score of each domain for all guidelines sorted by quality classification.

**Figure 3 nutrients-14-04579-f003:**
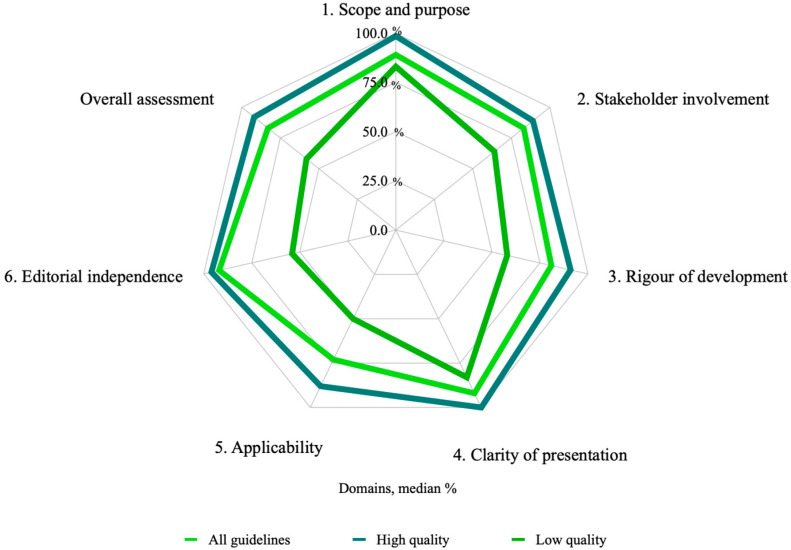
Median quality score of each domain for all guidelines sorted by quality classification.

**Table 1 nutrients-14-04579-t001:** Boolean search system.

(“Pregnancy”[Mesh] OR “Pregnancy Trimester, First”[Mesh] OR “Pregnancy Trimester, Second”[Mesh] OR “Pregnancy Trimester, Third”[Mesh]) AND (“Prenatal Care”[Mesh] OR “Nutrition Assessment”[Mesh] OR “Nutrition Therapy”[Mesh] OR “prevention and control” [Subheading] OR “Health Promotion”[Mesh]) AND (“Malnutrition”[Mesh] OR “Body Weight”[Mesh] OR “Anemia”[Mesh] OR “Deficiency Diseases”[Mesh] OR “Nutrition Disorders”[Mesh] OR “Gestational Weight Gain”[Mesh] OR “Fetal Growth Retardation”[Mesh] OR “Nutritional Physiological Phenomena”[Mesh])

MeSH: Medical subject headings.

**Table 2 nutrients-14-04579-t002:** General characteristics of the CPGs included in the analysis.

Reference	Clinical Guideline	Supporting Organization	Year of Publication	Region or Country	Number of References	Target Audience	Type of Recommendations
American College of Obstetricians and Gynecologists [19]	ACOG Practice Bulletin No 156: Obesity in Pregnancy	The American College of Obstetricians and Gynecologists	2015	USA	112	Obstetric care provider or healthcare professionals	P, D
American Diabetes Association [20]	Management of diabetes in pregnancy	American Diabetes Association.	2015	USA	23	Healthcare providers	P, D, T
ADA, ASN, Siega-Riz, A.M., King, J.C. [21]	Position of the American Dietetic Association and American Society for Nutrition: obesity, reproduction, and pregnancy outcomes	American Society for Nutrition	2009	USA	97	Registered dietitians, registered dietetic technicians, and other healthcare professionals	P, D, T
Australian Government Department of Health [22]	Clinical Practice Guidelines: Pregnancy Care	Australian Government Department of Health	2019	Australia	Not specified	Health professionals	P
Bomba-Opoń [23]	Folate supplementation during the preconception period, pregnancy, and puerperium. Polish Society of Gynecologists and Obstetricians Guidelines	Polish Society of Gynecologists and Obstetricians (PTGP):	2017	Poland	30	Not specified	P, T
Cetin, I. [24]	Breastfeeding during pregnancy: position paper of the Italian Society of Perinatal Medicine and the Task Force on Breastfeeding, Ministry of Health, Italy	Italian Society of Perinatal Medicine and Task Force on Breastfeeding	2014	Italy	66	Not specified	P, T
Charzewska. J. [25]	Prophylaxis of vitamin D deficiency, Polish Recommendations 2009	Not specified	2010	Poland	Not specified	Not specified	P, D, T
CMACE [26]	Management of Women with Obesity in Pregnancy	Royal College of Obstetricians and Gynecologists	2010	UK	77	Health professionals	P, D, T
Davies. G.A. [27]	SOGC Clinical Practice Guidelines: Obesity in pregnancy. No. 239, February 2010	Society of Obstetricians and Gynecologists of Canada	2010	Canada	79	Obstetrical care team	P, D, T
Davies. G.A.L. [28]	No. 239, Obesity in Pregnancy	The Society of Obstetricians and Gynecologists of Canada.	2018	Canada	78	Obstetrical care team	P, D, T
Denison, F.C. [29]	Care of Women with Obesity in Pregnancy: Green-Top Guideline No. 72	Royal College of Obstetricians and Gynecologists	2019	UK	202	Healthcare providers	P, D, T
Donnay, S. [30]	Iodine supplementation during pregnancy and lactation. Position statement of the working group on disorders related to iodine deficiency and thyroid dysfunction of the Spanish Society of Endocrinology and Nutrition	Spanish Society of Endocrinology and Nutrition	2014	Spain	73	Not specified	D
Fitzsimons, K.J. [31]	Setting maternity care standards for women with obesity in pregnancy	Centre for Maternal and Child Enquiries(CMACE)	2010	UK	55	Not specified	P, D, T
Fleming, N. [32]	Adolescent Pregnancy Guidelines	Canadian Pediatric and Adolescent Gynecology andObstetricians (CANPAGO)	2015	Canada	119	Healthcare providers	P, D
Haq, A. [33]	Clinical practice guidelines for vitamin D in the United Arab Emirates	Not specified	2018	United Arab Emirates	32	Physicians	D, T
Harden, C.L. [34]	Practice parameter update: management issues for women with epilepsy--focus on pregnancy (an evidence-based review): vitamin K, folic acid, blood levels, and breastfeeding: report of the Quality Standards Subcommittee and Therapeutics and Technology Assessment Subcommittee of the American Academy of Neurology and American Epilepsy Society	American Academy of Neurology	2009	USA	40	Not specified	P, T
Homer, C.S. [35]	Updated clinical practice guidelines on pregnancy care	Australian Health Ministers’ Conference and the Community and Disability Services Ministers’ Conference	2018	Australia	11	Health professionals	P, D
Jacob, C.M. [36]	Management of prepregnancy, pregnancy, and postpartum obesity from the FIGO Pregnancy and Non-Communicable Diseases Committee: A FIGO (International Federation of Gynecology and Obstetrics) guideline	International Federation of Gynecology and Obstetrics	2020	International	77	Healthcare practitioners	P, D, T
Koren, G. [37]	Cancer chemotherapy and pregnancy	Society of Obstetricians and Gynaecologists of Canada.	2013	Canada	88	Healthcare practitioners	P, T
Lausman, A. [38]	Intrauterine growth restriction: screening, diagnosis, and management	Society of Obstetricians and Gynaecologists of Canada	2013	Canada	55	Not specified	P, D, T
Marsh, J.C. [39]	Guidelines for the diagnosis and management of aplastic anaemia	General Haematology Task Force of the British Committee for Standards in Haematology	2009	UK	174	Health professionals	D, T
Maxwell, C. [40]	Guideline No. 391, Pregnancy and Maternal Obesity Part 1: Preconception and Prenatal Care	Society of Obstetricians and Gynaecologists of Canada (SOGC)	2019	Canada	178	Healthcare practitioners	P. T
Maxwell, C. [41]	Guideline No. 392, Pregnancy and Maternal Obesity Part 2: Team Planning for Delivery and Postpartum Care	Society of Obstetricians and Gynaecologists of Canada (SOGC)	2019	Canada	149	Healthcare practitioners	P, D T
Mottola, M.F. [42]	2019 Canadian guideline for physical activity throughout pregnancy	Society of Obstetricians and Gynaecologists of Canada’s (SOGC) Maternal Fetal Medicineand Guideline Management and Oversight Committees	2018	Canada	40	Obstetric care providers, policymakers, and fitness professionals	P
Muraro, A. [43]	EAACI food allergy and anaphylaxis guidelines. Primary prevention of food allergy	European Academy of Allergy and Clinical Immunology’s (EAACI)	2014	Europe	98	Medical secondary care, primary care, and nursing	P, T
NICE [44]	Antenatal care for uncomplicated pregnancies	National Institute for Health and Care Excellence: Guidelines	2019	UK	Not specified	Healthcare providers	P, D, T
NICE [45]	Diabetes in pregnancy: management from preconception to the postnatal period	National Institute for Health and Care Excellence: Guidelines	2020	UK	Not specified	Healthcare providers	P, D, T
NICE [46]	Weight management before, during, and after pregnancy	National Institute for Health and Care Excellence: Guidelines	2010	UK	15	Health professionals	P, D, T
Pavord, S. [47]	UK guidelines on the management of iron deficiency in pregnancy	British Committee for Standards in Haematology	2012	UK	81	Healthcare professionals	P, D, T
Piccinini, V.H. [48]	Canadian Adult Obesity Clinical Practice Guidelines: Weight Management Over the Reproductive Years for Adult Women Living with Obesity	The Canadian Association of Bariatric Physicians and Surgeons (CABPS)	2020	Canada	126	Primary care providers	P, T
Ryan, K. [49]	Significant haemoglobinopathies: guidelines for screening and diagnosis	British Society for Haematology	2010	UK	19	Not specified	D
Sentilhes, L. [50]	Shoulder dystocia: guidelines for clinical practice from the French College of Gynecologists and Obstetricians (CNGOF)	French College of Gynecologists and Obstetricians (CNGOF)	2016	France	11	Obstetricians or Health professional	P, T
Siu, A.L. [51]	Screening for Iron Deficiency Anemia and Iron Supplementation in Pregnant Women to Improve Maternal Health and Birth Outcomes: U.S. Preventive Services Task Force Recommendation Statement	U.S. Preventive Services Task Force (USPSTF)	2015	USA	27	Not specified	D, T
SMFM [52]	Society for Maternal–Fetal Medicine Consult Series #50: The role of activity restriction in obstetric management: (Replaces Consult Number 33, August 2014)	Society for Maternal–Fetal Medicine	2020	USA	47	Obstetricians and maternal–fetal medicinesubspecialists	T
WHO [53]	Proper maternal nutrition during pregnancy planning and pregnancy: a healthy start in life	Ministry of Health of Latvia and the WHO Regional Office for Europe	2017	Europe	27	Not specified	P
WHO [54]	Guideline: daily iron supplementation in adult women and adolescent girls	WHO Guidelines Approved by the Guidelines Review Committee	2016	International	28	Policymakers, expert advisers, and organizations involved in nutrition actions for public health	P
WHO [55]	Guideline: intermittent iron and folic acid supplementation in nonanaemic pregnant women	WHO Guidelines Approved by the Guidelines Review Committee	2011	International	32	Policymakers, expert advisers, and organizations involved in nutrition actions for public health	P
WHO [56]	Guideline: sodium intake for adults and children	WHO Guidelines Approved by the Guidelines Review Committee	2012	International	62	Policymakers, expert advisers, and organizations involved in nutrition actions for public health	P
WHO [57]	Guideline: sugar intake for adults and children, World Health Organization	WHO Guidelines Approved by the Guidelines Review Committee	2015	International	61	Policymakers, expert advisers, and organizations involved in nutrition actions for public health	P
WHO [58]	Guideline: vitamin A supplementation in pregnant women	WHO Guidelines Approved by the Guidelines Review Committee	2011	International	33	Policymakers, expert advisers, and organizations involved in nutrition actions for public health	P
WHO [59]	WHO antenatal care recommendations for a positive pregnancy experience: nutritional interventions update: multiple micronutrient supplements during pregnancy	WHO Guidelines Approved by the Guidelines Review Committee	2020	International	50	Policymakers, expert advisers, and organizations involved in nutrition actions for public health	P
WHO [60]	WHO antenatal care recommendations for a positive pregnancy experience: nutritional interventions update: vitamin D supplements during pregnancy	WHO Guidelines Approved by the Guidelines Review Committee	2020	International	56	Policymakers, expert advisers, and organizations involved in nutrition actions for public health	P
WHO [61]	WHO recommendations on antenatal care for a positive pregnancy experience	WHO Guidelines Approved by the Guidelines Review Committee	2016	International	214	Policymakers, expert advisers, and organizations involved in nutrition actions for public health	P, D

ADA: American Dietetic Association; ASN: American Society of Nutrition; CMACE: Centre for Maternal and Child Enquiries; SMFM: Society for Maternal–Fetal Medicine; NICE: National Institute of Health and Care Excellence; UK: United Kingdom; USA: United States of America; WHO: World Health OrganizationType of recommendations—P: prevention; D: diagnosis; T: treatment.

**Table 3 nutrients-14-04579-t003:** Appraisal of Guidelines for Research and Evaluation (AGREE) II version result for clinical practice guidelines.

Clinical Guideline	AGREE II Domains (%)							
	Scope and Purpose	Stakeholder Involvement	Rigour of Development	Clarity of Presentation	Applicability	Editorial Independence	Overall Assessment	Quality of Guidelines
12. Management of diabetes in pregnancy [20]	89	92	90	94	88	96	83	High quality
2019 Canadian guideline for physical activity throughout pregnancy [42]	100	97	92	100	96	88	92	High quality
ACOG Practice Bulletin No 156: Obesity in Pregnancy [19]	83	75	71	94	75	54	67	Low quality
Adolescent Pregnancy Guidelines [32]	97	86	81	100	67	75	83	High quality
Antenatal care for uncomplicated pregnancies [44]	100	97	100	100	94	92	100	High quality
Breastfeeding during pregnancy: position paper of the Italian Society of Perinatal Medicine and the Task Force on Breastfeeding, Ministry of Health, Italy [24]	83	69	57	25	38	88	58	Low quality
Canadian Adult Obesity Clinical Practice Guidelines: Weight Management Over the Reproductive Years for Adult Women Living with Obesity [48]	94	86	75	92	88	58	83	High quality
Cancer chemotherapy and pregnancy [37]	89	92	90	94	88	96	83	High quality
Care of Women with Obesity in Pregnancy: Green-top Guideline No. 72 [29]	86	58	97	97	65	96	83	High quality
Clinical practice guidelines for vitamin D in the United Arab Emirates [33]	94	72	51	89	73	29	67	Low quality
Clinical Practice Guidelines: Pregnancy Care [22]	69	56	56	83	48	71	58	Low quality
Diabetes in pregnancy: management from preconception to the postnatal period [45]	100	100	84	100	98	92	92	High quality
EAACI food allergy and anaphylaxis guidelines. Primary prevention of food allergy [43]	75	89	97	97	85	96	92	High quality
Folate supplementation during the preconception period, pregnancy and puerperium. Polish Society of Gynecologists and Obstetricians Guidelines [23]	39	28	21	64	4	8	17	Low quality
Guideline No. 391, Pregnancy and Maternal Obesity Part 1: Preconception and Prenatal Care [40]	97	83	100	94	60	83	83	High quality
Guideline No. 392, Pregnancy and Maternal Obesity Part 2: Team Planning for Delivery and Postpartum Care [41]	100	100	85	100	96	92	92	High quality
Guideline: daily iron supplementation in adult women and adolescent girls [54]	100	86	98	100	90	100	100	High quality
Guideline: intermittent iron and folic acid supplementation in nonanemic pregnant women [55]	100	83	96	92	100	83	92	High quality
Guideline: sodium intake for adults and children [56]	100	83	98	100	88	100	92	High quality
Guideline: sugars intake for adults and children. World Health Organization [57]	100	83	98	100	100	100	92	High quality
Guideline: vitamin A supplementation in pregnant women [58]	100	89	98	100	96	100	100	High quality
Guidelines for the diagnosis and management of aplastic anemia [39]	92	81	65	92	50	58	58	Low quality
Intrauterine growth restriction: screening, diagnosis, and management [38]	81	33	53	89	33	4	50	Low quality
Iodine supplementation during pregnancy and lactation. Position statement of the working group on disorders related to iodine deficiency and thyroid dysfunction of the Spanish Society of Endocrinology and Nutrition [30]	83	56	42	42	27	54	50	Low quality
Management of prepregnancy, pregnancy, and postpartum obesity from the FIGO Pregnancy and Non-Communicable Diseases Committee: A FIGO (International Federation of Gynecology and Obstetrics) guideline [36]	100	94	86	100	92	100	92	High quality
Management of Women with Obesity in Pregnancy [26]	100	89	92	94	83	92	67	Low quality
No. 239-Obesity in Pregnancy [28]	86	39	63	86	33	21	50	Low quality
Position of the American Dietetic Association and American Society for Nutrition: obesity, reproduction, and pregnancy outcomes [21]	75	47	22	56	25	29	25	Low quality
Practice parameter update: management issues for women with epilepsy—focus on pregnancy (an evidence-based review): vitamin K, folic acid, blood levels, and breastfeeding: report of the Quality Standards Subcommittee and Therapeutics and Technology Assessment Subcommittee of the American Academy of Neurology and American Epilepsy Society [34]	100	81	73	92	75	75	58	Low quality
Proper maternal nutrition during pregnancy planning and pregnancy: a healthy start in life [53]	89	81	63	83	63	96	67	Low quality
Prophylaxis of vitamin D deficiency: Polish Recommendations 2009 [25]	58	53	10	81	38	29	50	Low quality
Screening for Iron Deficiency Anemia and Iron Supplementation in Pregnant Women to Improve Maternal Health and Birth Outcomes: U.S. Preventive Services Task Force Recommendation Statement [51]	97	47	58	69	54	100	58	Low quality
Setting maternity care standards for women with obesity in pregnancy [31]	83	72	73	83	52	92	67	Low quality
Shoulder dystocia: guidelines for clinical practice from the French College of Gynecologists and Obstetricians (CNGOF) [50]	81	67	73	78	67	96	67	Low quality
Significant haemoglobinopathies: guidelines for screening and diagnosis [49]	81	92	58	89	58	8	50	Low quality
Society for Maternal–Fetal Medicine Consult Series #50: The role of activity restriction in obstetric management: (Replaces Consult Number 33, August 2014) [52]	75	44	53	72	42	50	58	Low quality
SOGC Clinical Practice Guidelines: Obesity in Pregnancy. No. 239, February 2010 [27]	83	39	63	81	31	38	50	Low quality
UK guidelines on the management of iron deficiency in pregnancy [47]	75	64	67	78	54	100	67	Low quality
Updated clinical practice guidelines on pregnancy care [35]	97	94	90	100	63	88	83	High quality
Weight management before, during and after pregnancy [46]	100	100	92	100	94	100	92	High quality
WHO antenatal care recommendations for a positive pregnancy experience: nutritional interventions update: multiple micronutrient supplements during pregnancy [59]	89	86	90	86	75	100	83	High quality
WHO antenatal care recommendations for a positive pregnancy experience: nutritional interventions update: vitamin D supplements during pregnancy [60]	89	89	90	92	85	100	92	High quality
WHO recommendations on antenatal care for a positive pregnancy experience [61]	89	92	88	92	85	100	92	High quality
Mean (range)	88.3 (range 39 to 100%)	75.4 (range 28 to 100%)	74.3 (range 10 to 100%)	87.2 (range 25 to 100%)	67.8 (range 4 to 100%)	75 (range 4 to 100%)	72.9 (range 17 to 100%)	
Median (range)	89 (range 39 to 100%)	83 (range 28 to 100%)	81 (range 10 to 100%)	92 (range 25 to 100%)	73 (range 4 to 100%)	92 (range 4 to 100%)	83 (range 17 to 100%)	

EAACI: European Academy of Allergy and Clinical Immunology; WHO: World Health Organization.

## Data Availability

Not applicable.

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
