# Peer review of "Quality Appraisal of Nutritional Guidelines to Prevent, Diagnose, and Treat Malnutrition in All Its Forms during Pregnancy"

_nutrients, 2022, doi:10.3390/nu14214579_

Round 1

Reviewer 1 Report

The current study is a systematic review which performs a quality appraisal on guidelines regarding the maternal nutrition in pregnancy. The authors have used scientifically correct the AGREE II tool for appraisal of the guidelines.It is a good and precise study. My question is the following and it is related with the flowchart . Which guidelines of study selection and flowchart exhibition follows the flowchart you have used? Is it PRISMA? Please clarify this. Otherwise it is a good acceptable paper.

Reviewer 2 Report

This paper presents results from a quality assessment of CPGs related to prevention, diagnosis, and treatment of malnutrition during pregnancy. The review seems to be comprehensive, and authors used a standardized assessment tool (the AGREE II) in their quality appraisals. I think the assessments provided in the paper are potentially helpful, but the authors can do a better job (a) describing their methodology and (b) being more specific in the discussion of their findings. I describe specific ways/areas in which these two elements can be improved, below, along with other specific comments. Also, the paper needs to be thoroughly edited for English language usage and typos. I note some of these issues below, but I did not note all of them. Finally, the authors should review the paper to ensure the tense is consistent throughout (it currently switches between past and present tense).

Specific comments:

Abstract: Small point, but I don’t think the use of the semicolon in the first sentence is grammatically correct. Semi-colons should separate two complete sentences, and the phrase after the semicolon is not a complete sentence. Could drop the semicolon and replace “furthermore, and…” with “and to…”

Abstract: Here and throughout the paper, it is not really clear what is meant by assessing the “methodological quality” of these CPGs. It seems like many of the domains are not methodological domains, per say. Perhaps consider dropping “methodological” and just refer to assessing the quality of the CPGs?

Abstract: “Score” should rather be “scoring”?

Abstract: Instead of “The high-quality CPGs,…” should rather be something like “Among the high-quality CPGs…”

Introduction, 5th paragraph: Not clear why the second statement begins with “On the other hand.” Seems that the evidence from the systematic reviews and observational studies are both suggesting the important of nutrition.

Introduction, last paragraph: Again, not really clear what is meant by methodological quality.

Study design and eligibility criteria, first paragraph: After reading through the methods section, I’m not clear what the five stages of the review process were.

Search strategy and studies selection section: Frame time should rather be time frame.

Quality assessment, paragraph about the use of the likert scale: It is not clear from this description what “strongly agree” and “strongly disagree” are referring to. Agree and disagree with what, exactly?

Quality assessment: How was the threshold of 70% for “high quality” determined? Is this indicated in the AGREE User’s manual/the standard way the scores are categorized? If not, please describe how/why this specific threshold was adopted.

Quality assessment, last paragraph: When you say “Two authors (BSM, LTC, NRM, SES, AT, FAA, JMC, LIG, and MAM) independently evaluated the quality of each CPGs using the online AGREE platform, "My AGREE PLUS.", how were the two scores combined? An average? What if the two scores were quite different?  Also, is it standard practice to have just two people independently review? Seems like basing scores on the assessments of just two people, which will undoubtedly include some judgement calls, may be too few? Seems like it would have been better to have each CPG reviewed by more people, especially since you paper calls out several CPGs as very poor quality.

Quality assessment: Overall, your methods here need to be more fully described. In addition to the two questions I raised above about the quality assessment, it is also not clear how you arrived at the “overall assessment” score – it seems like it might be logical for this to be an average of the scores in each of the domains, but it doesn’t appear that is the case. Also, how was it determined that categorization of high quality should be based on four or more domains reaching a score higher than 70% rather than looking at the overall assessment score (also, as I note below, this methodological choice is not described in the methods section but should be). It should also be made clear whether the procedures you followed to come up with your overall assessment score and for categorizing the studies as high or low quality were defined as part of the AGREE process or whether these were something you came up with yourselves (and if so, how). Also, a sentence or two summarizing the overall objective of the AGREE framework would be helpful.

Data analysis: Not clear how means and medians would allow you to identify the most critical (i.e. the most important) domains. Do you rather mean to identify the domains with the lowest and highest average quality, or something like that?

Table 2. I’m guessing that the reference “12. Management of diabetes in pregnancy” should not have the 12 in front of the title?

Section 3.1: You note here that CPGs were classified as high quality if four or more domains reached a score higher than 70%. This should also be noted in the methods section. And please also explain how the four-or-more criteria was arrived at. Is this the AGREE standard? If not, please describe how/why you applied this specific criteria.

3.1.1. Scope and Purpose Domain: Should rather be issued to Bomba-Opoń rather than issued by Bomba-Opoń, right, as issued by would imply that Bomba-Opon gave themselves the score?

Discussion, second paragraph: I think this paragraph (and/or the paragraphs that follow this paragraph) can be expanded to be more useful by providing some additional details about what made the CPGs relatively good in the domains of scope and purpose and clarity of presentation and, more importantly, relatively poor in the domains of applicability, rigor of development, and editorial independence. That is, in what ways (described summarily but specifically) did this collection of CPGs do a good job meeting the criteria of the generally highly scored domains and what were the shortcomings that led to them not meeting the criteria in other domains (i.e., among the low-scoring domains, be more specific about what elements were lacking in this collection of CPGs and/or how, specifically, they could be improved). It would be helpful if these additional details spoke specifically to improvements needed in these guidelines to improve the recommendations for preventing, diagnosing, and treating women's malnutrition during pregnancy (not just for all guidelines in general).

Round 2

Reviewer 2 Report

The authors have done a nice job revising the paper. It still has some typos and grammatical errors, which should be corrected before publication. I've listed a few below.

Abstract: You implemented the suggestion to replace score with scoring in one place, but you also replaced score with scoring here, which is not grammatically correct: “The two domains that obtained the highest scoring were Scope and purpose with 88.3%...”

p. 3, Study design and eligibility criteria section: Step iv should rather be “Charting the data”

 p. 4, second sentence: Should be “This instrument aims to” rather than “This instrument aim to”

 p. 4, results: Typo. You say: “We identify 82 records through the search.” but this should rather be “We identified 82 records through the search.”